# Current Status and Future Perspectives of Androgen Receptor Inhibition Therapy for Prostate Cancer: A Comprehensive Review

**DOI:** 10.3390/biom11040492

**Published:** 2021-03-25

**Authors:** Tae Jin Kim, Young Hwa Lee, Kyo Chul Koo

**Affiliations:** 1CHA Bundang Medical Center, Department of Urology, CHA University College of Medicine, Seongnam 13496, Korea; tjkim81@cha.ac.kr; 2Department of Urology, Gangnam Severance Hospital, Yonsei University College of Medicine, Seoul 06229, Korea; wjungyh@naver.com

**Keywords:** androgen deprivation therapy, androgen receptor, androgen receptor inhibition, prostate cancer

## Abstract

The androgen receptor (AR) is one of the main components in the development and progression of prostate cancer (PCa), and treatment strategies are mostly directed toward manipulation of the AR pathway. In the metastatic setting, androgen deprivation therapy (ADT) is the foundation of treatment in patients with hormone-sensitive prostate cancer (HSPC). However, treatment response is short-lived, and the majority of patients ultimately progress to castration-resistant prostate cancer (CRPC). Surmountable data from clinical trials have shown that the maintenance of AR signaling in the castration environment is accountable for disease progression. Study results indicate multiple factors and survival pathways involved in PCa. Based on these findings, the alternative molecular pathways involved in PCa progression can be manipulated to improve current regimens and develop novel treatment modalities in the management of CRPC. In this review, the interaction between AR signaling and other molecular pathways involved in tumor pathogenesis and its clinical implications in metastasis and advanced disease will be discussed, along with a thorough overview of current and ongoing novel treatments for AR signaling inhibition.

## 1. Introduction 

Prostate cancer (PCa) is the most frequently diagnosed neoplasm in developed countries and is accountable for the second-highest rate of cancer-related deaths [1]. Significant advances in treatment modalities and novel clinical strategies for the treatment of PCa have been developed in the last decade. The androgen receptor (AR) has been generally regarded as the most integral factor for regulating the development and spread of tumor cells, and consequently, the majority of treatment regimens are directed against the AR pathway. For metastatic PCa or in the advanced setting, androgen deprivation therapy (ADT) is the treatment of choice for hormone-sensitive disease. Results from recent clinical trials support the additional use of docetaxel, abiraterone, enzalutamide, or apalutamide with ADT in the treatment of metastatic hormone-sensitive PCa (mHSPC) [2,3,4,5,6,7,8]. However, this transient hormone-sensitive phase lasts for a median of 18 to 36 months, and the majority of patients will progress to metastatic castration-resistant PCa (mCRPC) [9]. Treatment modalities for mCRPC patients include chemotherapy agents, such as docetaxel and cabazitazel, androgen receptor axis targeted (ARAT) agents, such as abiraterone acetate and enzalutamide, radiotherapy with radium-223, and immunotherapeutic approaches using sipuleucel-T [5,8,10,11,12,13,14,15]. However, even with these improvements and novel therapeutic strategies, mCRPC has a poor clinical prognosis and outcome [16]. 

The underlying molecular mechanisms that lead to progression to CRPC are associated with the conservation of AR signaling in the castration-level androgen setting [17]. Clinical findings in recent studies have revealed various growth-promoting and survival pathways in the carcinogenesis of PCa, which implies the importance of understanding the alternative cellular pathways in disease progression and the interaction of these pathways with AR signaling. Encouraging therapeutic outcomes of targeted agents for a specifically defined molecular subpopulation in other solid cancers provides a strong clinical rationale for improving the treatment strategies in patients with castration-resistant prostate cancer (CRPC), utilizing specifically tailored clinical regimens. 

The need to understand the intricate relationship between AR signaling and other molecular cascades involved with the pathogenesis of PCa and its therapeutic implications in advanced disease is of utmost importance, along with the familiarity of current and novel therapeutic approaches utilizing the AR signaling pathway for the treatment and management for patients with PCa in the advanced setting.

## 2. The Androgen Receptor and Tumor Microenvironment in Prostate Cancer 

Classified as a steroid hormone receptor, the AR is a ligand-activated nuclear transcription factor whose role is the regulation of target gene expression. The AR consists of a DNA-binding domain (DBD), along with a hinge region and ligand-binding domain (LBD) and an N-terminal domain (NTD) that consists of two repeated polymorphic trinucleotide segments. The repeat segments are constituted of random polyglutamine and polyglycine repeats, which are involved in the regulation of AR transcriptional activity [18,19]. AR expression is found in most primary and metastatic PCa patients of every stage and grade, and these characteristics of AR expression are observed in most cases of CRPC. AR signaling is constantly active and provides the basis for the progression and survival of prostate tumor cells [20,21]. 

The prostate tumor microenvironment (TME) harbors a various number of inflammatory cells and stromal components. Of these cellular constituents, cancer-associated fibroblasts (CAFs) are recruited and activated by factors secreted from the primary cancer, enabling molecular crosstalk with tumor cells. This interaction activates CAFs, which initiate cytokine release and extracellular matrix (ECM) deposition. This molecular cascade results in the reorganization of the structure and composition of the connective tissue along with the release of growth factors that influence the modification of the ECM. This alteration increases tumor stiffness and induces the proliferation and growth, along with drug-resistance of the newly transformed epithelial cells [22]. The CAFs of the prostate enhance the transformation of the gland and stimulate cancer progression [23,24,25]. Moreover, these fibroblasts form a cell cluster that retains cancer stem cell (CSC) abilities, which facilitate the metabolic alterations of the PCa cells or their epithelial–mesenchymal transition (EMT) and the progression of metastatic disease [26,27,28,29].

In the TME, tumor-associated macrophages (TAMs) comprise the majority of the population within the cellular milieu and are usually activated by a variety of chemokines, from chemokine C-C motif ligand 2 (CCL2) to CCL5 [30]. Dependent on its complementary mediators, the cellular characteristics of TAMs can be a regulator for tumor proliferation and development, or be associated with a good disease prognosis. A recent study has shown that TAMs stimulate CCL2, which induces a signal transducer and activator of transcription 3 (STAT3), a mediated EMT cascade [31]. Activation of EMT is considered as major contributing factor that alters the function of immune cells in the TME, leading to immune resistance and suppression of the immune system [31]. Various clinical studies have confirmed the pro-tumorigenic effects of TAMs that can be utilized as a predictive biomarker of clinical outcome in patients with PCa [32,33,34].

The molecular crosstalk and relationship between the prostate TME and AR signaling are of an intricate nature, in which the AR and TME have both tumorigenic and anti-tumorigenic roles [35]. The AR is expressed by the stromal cells, which are crucial in glandular development and growth of the prostate and are also a crucial factor in prostate carcinogenesis. Studies have reported that the expression of AR in the prostate epithelium is imperceptible during prostatic growth and development, while high levels of AR are expressed in the stromal cells [25,36]. In contrast to the AR expression levels in the setting of PCa, studies have shown that low AR expression in the stromal cells is related to biochemical recurrence (BCR), high risk of disease progression, and worse clinical outcomes [29,37].

As the benign prostate cells transform into cancer cells, the structural and genomic transition of the stromal cells results in a progressive decrease in AR expression. Starting from low-grade PCa, there is a linear decrease in the mean AR stromal expression, which is almost entirely absent in high-grade disease [38]. It has been shown that the mean AR expression is decreased both in the epithelium and adjacent stroma along with tissue dedifferentiation, while the depletion is more pronounced in the stromal nuclei [38]. In the disease spectrum of PCa, patients with metastatic disease have shown decreased AR stromal expression compared to that in the corresponding primary prostate tumor [39]. Moreover, expression levels in the stromal cells are distinctly decreased in patients with CRPC when compared with patients with hormone-sensitive PCa (HSPC) [39]. Further studies are warranted to elucidate the underlying pathophysiology regarding the loss of AR expression in stromal cells. However, studies have reported that the AR signaling in stromal cells and epithelial cells target different individual genes. Moreover, the molecular transition from an androgen-dependent paracrine pathway to an autocrine pathway during prostate carcinogenesis ultimately results in the molecular independence of cancer cells from the stromal–epithelial interactions for cellular progression and proliferation [40,41].

## 3. Androgen Receptor Resistance Mechanisms in Treatments for Advanced and Metastatic Prostate Cancer

According to the study results of Abeshouse et al., AR gene mutations and amplifications are observed in 1% of patients with PCa [42], and approximately 60% of the patients with metastatic tumors harbor AR gene mutations and amplification [43,44,45]. In contrast to patients with PCa who underwent ADT monotherapy, those with metastatic PCa who were administered AR antagonist regimens had higher incidences of AR mutations [46,47]. Most AR mutations are limited to the AR’s androgen-binding domain boundaries, ultimately resulting in single-amino-acid substitutions [46,47]. These mutations functionally enable anti-androgens to perform as AR agonists, providing a significant molecular benefit for cancer development and disease progression. AR mutations develop throughout the course of disease progression and depend on the extent of the metastasis, while the mutation levels are influenced by ADT administration and other treatment modalities [48]. Various clinical studies have described over 600 mutations in the AR with a mutational frequency of <25% in androgen-dependent PCa and a mutation frequency rate of more than 50% in patients with androgen-independent and metastatic PCa [49]. The majority of AR mutations are gain-of-function point mutations [50] and the most frequently noted mutation is T877A, which is observed in roughly 30% of mCRPC patients [45,47,51]. This type of mutation facilitates AR activation by other adrenal secreted androgens, which include progesterone [52,53], dehydroepiandrosterone (DHEA) [54], and androstenediol [55]. This molecular alteration allows continued expression of AR despite castration levels of androgens and may be a potential reason for the low response rate of castration therapy in patients with AR mutation-expressing tumors. Other underlying resistance pathophysiologies to ADT include AR gene amplification, increased production of steroids by tumor cells and higher androgen uptake on the cellular level, AR receptor promiscuity, and increased rates of 5α-reductase expression [56,57,58]. These mechanisms of resistance cause excessive AR transcription or higher expression of the ligands that result in constant stimulation and activation of the LBD. ADT monotherapy has a minimal therapeutic effect on the resulting increase in AR transcriptional activity. In a study led by Mononen et al., results revealed that other germline polymorphisms in the AR are associated with increased risk of PCa occurrence [59]. H874Y and W435L mutations cause increased co-regulator binding of AR, subsequently leading to increased AR transcription [60,61]. Furthermore, studies of other mutations, such as F876L, have observed mutation-induced resistance to enzalutamide and apalutamide [62,63]. A tendency toward increased AR amplifications is found in patients who underwent PCa treatment and in CRPC patients when compared to treatment-naïve primary PCa. The wild-type sequence amplification enhances the sensitivity of prostate cells to the decreased castration levels of androgens, which allows the prostate tumor cells to persist in a low androgen level environment [64,65].

As a therapeutic form of castration, ADT plays an essential role in the management of metastatic PCa and patients in the advanced setting. This chemical regimen inhibits the production of androgens and consists of anti-androgens that competitively bind to the C-terminal of the AR, resulting in activation blockage of the AR. The ADT regimen usually consists of gonadotropin-releasing hormone (GnRH) agonists that inhibit the production of androgens, such as leuprolide acetate, triptorelin, goserelin acetate, and histrelin [66,67,68]. After the initial administration of GnRH agonists, there is a brief increase in the testicular production of androgens [58]. Despite this sudden surge of testosterone, the constant stimulation of pituitary GnRH receptors eventually results in the downregulation of androgens and AR desensitization. This suppresses luteinizing hormone (LH) release and accomplishes an overall reduction in the circulating levels of testosterone, estrogen, and progesterone [58]. GnRH antagonists are used in patients where the testosterone “flare” is undesirable. Although GnRH antagonists have a different mechanism, the therapeutic objective of testosterone reduction is the same. By reversibly inhibiting GnRH receptors on the anterior pituitary gland, the administration of GnRH antagonists ultimately results in the blockage of LH secretion [69]. Although the toxicities of ketoconazole overshadow its therapeutic advantages, adrenal androgen inhibitors with the combined administration of corticosteroids, ketoconazole, and aminoglutethimide can be administered for the complete blockade of testosterone production [70]. Even with ADT monotherapy, the majority of patients with PCa will develop resistance to the regimen and progress to CRPC within 18 months to 3 years [58,71,72,73,74]. In addition to the previously mentioned therapeutic regimens, the first generation of anti-androgens with a different mode of action has been developed. These anti-androgens, including bicalutamide, nilutamide, and flutamide, activate co-repressors and inhibit co-activators to suppress the AR transcription cascade while allowing the AR to enter the nucleus and bind to the designated deoxyribonucleic acid (DNA) [75,76]. 

Recent studies and clinical trials have focused on other pathways of AR alterations to better comprehend the vast clinical spectrum of therapeutic resistance to ADT in the clinical setting of PCa. Several AR variants (AR-Vs) that result in natural resistance have been observed in CRPC cell lines [77]. Most AR-Vs in the clinical setting have an absent ligand-binding domain while preserving their DNA binding activity in an androgen-absent environment and display consistent molecular activity. AR-V7 is the only variant identified at the protein level and is constantly expressed in tissue specimens of recurrent PCa and in cell lines of CRPC patients [78,79,80]. Another source for AR variants is exon skipping during messenger ribonucleic acid (mRNA) splicing. For instance, ARv567es expresses the C terminal of the LBD, encoded by exon 8, but does not express exons 5, 6, and 7, which are crucial for coding a portion of the LBD. Moreover, ARv567es shares similar properties with AR-V7 in the sense that it is constitutively active and can undergo protein translation [81]. Various studies have confirmed the presence of ARv567es expression in blood samples of PCa patients; therefore, ARv567es upregulation is considered a potential mechanism of resistance in the AR variant-dependent setting [81,82,83,84,85]. Unlike a full-length AR, both AR-V7 and ARv567es variants exhibit dissimilar transcriptional activity. Clinical data from Hu et al. showed that AR-V7 has gene expression abilities that are related to cell cycle progression [86], and further studies showed that the UBE2C gene is a major component in this process by regulating the gene expression associated with the G1/Sand G2/M transition and M phase of the cell cycle [87]. In addition, both AR variants mentioned above have been discovered in benign prostate hyperplasia (BPH) and CRPC samples. However, AR-V7 was the only variant detected in hormone-naïve PCa [70,81]. The fact that AR-V7 overexpression is related to increased risks of BCR after radical prostatectomy (RP) in patients with HSPC and with worse outcomes in patients with CRPC provides validation for the potential usage of AR-V7 as a prognostic biomarker [69,82].

## 4. Novel Androgen Receptor Inhibitors in the Treatment of Non-Metastatic Castration-Resistant Prostate Cancer

Even in the setting of CRPC, AR signaling continues as a significant factor of tumor growth and disease development [88]. The promising results of clinical trials based on androgen receptor signaling inhibitors (ARSi), such as enzalutamide, darolutamide, and apalutamide, have provided the rationale for using ARSi as a treatment of choice for CRPC patients [89]. These AR antagonists block AR nuclear translocation and reduce the expression of androgen-dependent genes due to higher rates of bonding to the AR binding domain and no agonist properties [90]. In the AFFIRM trial (NCT00974311), a phase III double-blind clinical study, enzalutamide significantly improved overall survival (OS) in CRPC patients who underwent prior chemotherapy compared to placebo [8], and results of the phase III double-blind PREVAIL study (NCT01212991) showed that administration of enzalutamide significantly decreased the risk of disease progression in chemotherapy-naïve settings [10]. 

In the double-blinded, randomized phase III PROSPER trial (NCT02003924), 1401 patients with non-metastatic CRPC with increasing prostate-specific antigen (PSA) levels were administered daily with 160 mg of enzalutamide or placebo. The results showed that combining enzalutamide with ADT improved median OS to 67.0 months compared with the placebo plus ADT group, which was 56.3 months (HR 0.73, 95% CI 0.61–0.89; *p* = 0.001). The most common adverse events (AE) were fatigue and musculoskeletal events, which occurred in 31% of the enzalutamide arm in comparison to 21% in the placebo group [91].

Founded on the promising results of phase III clinical studies, apalutamide and darolutamide have both gained approval from the Food and Drug Administration (FDA) for treatment in the clinical setting of non-metastatic CRPC. In the Selective Prostate Androgen Receptor Targeting with ARN-509 (SPARTAN) study (NCT01946204), 1207 patients were randomly allocated to receive a daily dose of 240 mg of apalutamide or placebo [92]. The study concluded that metastasis-free survival (MFS) was increased by two years in the apalutamide group (40.5 vs. 16.2 months), and the time to symptomatic progression was significantly longer in the apalutamide group (HR 0.45, 95% CI 0.32–0.61; *p* < 0.001) [92,93].

The phase III Androgen Receptor Antagonizing Agent for Metastasis free survival (ARAMIS) clinical trial (NCT02200614) was designed to analyze the therapeutic effects of the combination of darolutamide and ADT compared to ADT monotherapy. A total of 1509 non-metastatic CRPC patients were assigned in random to ADT with either darolutamide or placebo [94]. After a median follow-up of 29 months, the final analysis reported a statistically significant OS improvement (81% vs. 77%, respectively) while the treatment cohort showed a reduction of 31% in the risk of death (HR 0.69, 95% CI 0.53–0.88, *p* = 0.003) [95]. Darolutamide administration also showed therapeutic advantages in all secondary and exploratory endpoints in the intent-to-treat population (ITT), including time to pain progression (*p* < 0.001), delay of initial chemotherapy (*p* < 0.001), and the first onset of a symptomatic skeletal event (*p* = 0.005). Adverse events of darolutamide, which consisted of fatigue (12.1% vs. 8.7%), back pain (8.8% vs. 9.0%), arthralgia (8.1% vs. 9.2%), and hypertension (6.6% vs. 5.2%), were dissimilar between the study groups [95]. 

According to the aforementioned clinical trials, ARSi agents clearly offer statistically valid and therapeutic value in the ongoing clinical management of non-metastatic CRPC. Studies have shown that the occurrence rate of adverse events to darolutamide is lower in comparison to other AR inhibitors [96]. Given its favorable toxicity profile, darolutamide shows potential to be the treatment of choice for patients with a neurological medical history or a high risk for neurologic side effects. The continuing phase II clinical trials ODENZA (NCT03314324) [97] and Androgen Receptor Directed Therapy on Cognitive Function in Patients treated with Darolutamide or Enzalutamide (ARACOG) (NCT04335682) [98] will compare enzalutamide and darolutamide in the metastatic CRPC setting and will explore the outcome results of patient preference and cognitive function, respectively. Clinical data from the previously mentioned clinical trials have shown a significant link between MFS and OS, which provided evidence for MFS to be a clinically important surrogate endpoint, resulting in the approval of these three AR antagonists as treatment modalities for non-metastatic CRPC [99,100]. In comparison to ADT monotherapy, recent results and final analyses of the PROSPER, SPARTAN, and ARAMIS trials showed statistically significant improvement in OS for patients with non-metastatic CRPC who were randomly assigned to enzalutamide, apalutamide, and darolutamide, respectively. [91,95,101]. Table 1 summarizes the clinical trials evaluating the clinical benefits of ARSi in the treatment of non-metastatic CRPC. 

## 5. Androgen Receptor Inhibition in Metastatic Hormone-Sensitive Prostate Cancer

In the phase III double-blind LATITUDE trial (NCT01715285) [5], the therapeutic efficacy of the combined administration of abiraterone acetate with prednisone plus ADT was compared to ADT with placebo. A total of 1209 mHSPC patients with at least two out of three risk factors ascertaining an unfavorable prognosis, which included a Gleason score of 8 or above, three or more bone metastatic sites identified by bone scan, or visceral metastases, were recruited. A final number of 1199 patients were randomly allocated to abiraterone acetate and prednisone with ADT (abiraterone group) or to ADT with placebo. After a median follow-up of 30 months, the abiraterone group exhibited a survival benefit in terms of the median OS, which was not reached while the OS of the placebo group was 34 months (HR 0.62, 95% CI 0.51–0.76; *p* < 0.001), and a radiographic PFS (rPFS) of 33 months for the abiraterone cohort was observed compared to a 14 months rPFS for the placebo arm (HR 0.47, 95% CI 0.49–0.55; *p* < 0.001) [5]. After a median follow-up of 51 months, the final outcomes of the trial confirmed the clinical benefits of combined abiraterone acetate administration with ADT and prednisone. The results showed an OS of 53 months for the abiraterone cohort compared to an OS of 36 months for the placebo arm, along with a 34% risk reduction (*p* < 0.001). Excluding patients with an Eastern Cooperative Oncology Group (ECOG) performance status of 2, an OS benefit was observed in all subgroups of the trial, and improvements were observed in all secondary endpoints in patients who were prescribed the study drug. From the promising results of this clinical trial, abiraterone acetate plus prednisone has become a new treatment choice for patients with mHSPC [5].

The ARCHES trial (A Randomized, Phase III Study of Androgen Deprivation Therapy With Enzalutamide or Placebo in Men With Metastatic Hormone-Sensitive Prostate Cancer) (NCT02677896) was a phase III study that analyzed the safety profile and therapeutic efficacy of enzalutamide. The study cohort consisted of 1150 patients with mHSPC, stratified according to prior docetaxel chemotherapy and volume of the disease [2]. Patients were randomly assigned to enzalutamide and ADT combination administration or placebo with ADT. In comparison to the placebo arm, the enzalutamide plus ADT regimen arm exhibited a reduced risk of rPFS or death by 61% (*p* < 0.001). The therapeutic advantages of enzalutamide were observed in all specified subgroups, which consisted of patients with low-volume disease and patients who had received prior chemotherapy with docetaxel. Incidence of grade 3 or higher AE was comparable between the study groups at 24% for enzalutamide-treated patients and 25% for the placebo group. The study concluded that enzalutamide treatment significantly reduced metastatic progression or death, and due to its safety profile, enzalutamide should be considered a viable therapeutic option for patients with mHSPC [2].

Results of the phase III, randomized, open-label trial ENZAMET (Enzalutamide in First-Line Androgen Deprivation Therapy for Metastatic Prostate Cancer) (NCT02446405) [4] verified a significant OS improvement with the treatment regimen of enzalutamide at a daily dose of 160 mg with concurrent ADT when compared to standard ADT monotherapy with or without docetaxel chemotherapy. Moreover, the enzalutamide group showed better PSA progression-free survival (HR, 0.39; *p* < 0.001) and PFS (HR, 0.40; *p* < 0.001) [4].

The therapeutic benefit of apalutamide was explored in the phase III, double-blind, TITAN (The Targeted Investigational Treatment Analysis of Novel Anti-androgen) (NCT02489318) clinical trial [3]. In this study, 525 patients with mHSPC were allocated to receive apalutamide at 240 mg a day, while 527 patients were given placebo while undergoing ADT. This trial reported that apalutamide significantly improved the OS at 24 months (82.4% in the apalutamide group vs. 73.5% in the placebo group; HR, 0.67; 95% CI, 0.51–0.89; *p* = 0.005) and rPFS at 24 months was 68.2% in the study drug cohort while 47.5% was noted in the placebo group. Grade 3 or 4 adverse events were observed with comparable frequency across the two study populations at 42.2% in the apalutamide group and 40.8% in the control group [3]. Table 2 depicts the current and ongoing trials investigating the efficacy of AR inhibition in the mHSPC setting. 

## 6. Selective Androgen Receptor Degraders and Proteolysis Targeting Chimeras 

The expression of AR is crucial in the development and progression of PCa, and for this reason, ADT is considered the mainstream treatment of PCa. When stimulated, AR translocation to the nucleus is initiated, and regulation of the genetic code needed for cell growth and proliferation occurs [102]. The purpose of ADT is to inhibit testosterone production and therefore diminish AR stimulation. However, due to the limited therapeutic efficacy of ADT, the majority of patients will ultimately progress to CRPC [103]. Recent studies have shown that AR inhibition can be approached in a different setting by signal blocking or by degrading the AR [104]. A novel strategy targeting AR degradation has shown potential as an alternative approach in patients with mCRPC. Selective AR degraders (SARD) and proteolysis-targeting chimeras (PROTAC) are two different molecular structures that have demonstrated antitumor properties in PCa patients by AR degradation. Initial studies done by Bradbury et al. showed that SARDs had moderate AR downregulation properties in PCa cells, but that this therapeutic effect was overshadowed by adverse cardiovascular effects [105]. However, further research and development of SARDs led to the manufacture of molecular compounds with minimum cardiotoxic side effects. In a phase I clinical study done by Omlin and colleagues, the study drug AZD3514 demonstrated a PSA > 50% decline in 13% of the study population and objective soft tissue responses in 17% of individuals [106]. Recent study results based on ASC-J9 showed that the SARD demonstrated AR degradation in both wild-type AR and AR splice variants in ex vivo models [107]. Further preclinical studies showed that ASC-J9 was effective against AR mutants in enzalutamide-resistant CRPC and counteracted AR enhancement in CRPC cells exposed to docetaxel [108,109]. Moreover, a novel class of SARDs tagged with hydrophobic residue, which mimics a partial protein denaturation and activates proteasome-mediated AR degradation, has been shown to overcome enzalutamide resistance in ex vivo settings [110].

Proteolysis-targeting chimeras (PROTACs) are molecular structures that consist of a trimeric complex between a target protein and an E3 ubiquitin ligase, which enables ubiquitination of the target followed by consequent AR degradation [111,112]. Recent data gathered on ARCC-4, a low-nanomolar AR degrader, shows this particular PROTAC has an approximately 95% ability in cellular AR degradation [112]. Moreover, various studies have shown that, even in a high androgen milieu, ARCC-4 inhibits prostate tumor cell proliferation and degrades enzalutamide-resistant AR with point mutations, which makes this PROTAC a potential therapeutic option in enzalutamide-resistant patients [113]. In addition, in vitro and in vivo results of ARD-61 in enzalutamide-resistant models have shown promising anti-proliferative and tumor apoptotic activity [114]. 

In a phase I clinical trial of ARV-110 (NCT03888612), an orally bioavailable PROTAC, 18 patients with at least two prior therapies regarding mCRPC were administered 140 mg of ARV-110 daily. The study reported that two patients achieved a reduction in PSA of ≥50%. Two patients developed grade 3/4 elevated hepatic enzyme levels, but the overall safety profile of the drug was tolerable [115]. Although second-generation AR inhibitors have an influential role in the treatment of PCa patients with metastasis or advanced disease, constant clinical research and development of new generation AR degraders and alternative therapeutic modalities will broaden the treatment modalities in the disease continuum of PCa. Ongoing studies and clinical trials based on SARDs and PROTACs are shown in Table 3. 

## 7. Future Directions and Perspectives in Androgen Receptor Inhibitor Therapy

With the continuous surge in the incidence of PCa, constant improvement in diagnostic modalities and therapeutic options in the clinical setting are mandatory. The significance of AR signaling in the disease continuum of advanced PCa and mCRPC certifies that the AR receptor will remain as the primary therapeutic target. Despite robust clinical research and developments in the pharmaceutical industry for novel anti-androgens, unmet clinical needs remain, and the various mechanisms of resistance that lead to CRPC suggest the requirement of a fundamentally distinct therapeutic approach with novel drug targets. 

Numerous mutations of the AR have been described as the underlying mechanism to treatment failure, which opens the opportunity for advances and improvements in the diagnosis of PCa at the molecular level. The clinical nature of PCa mandates the importance of identifying these mutations for a tailored-administration of androgen inhibitors most suitable to each patient. Due to its key role in the transcription process, the AR-NTD has become a target for potential novel inhibitors. Although it is a significant challenge, targeting AR degradation or signal blockage provides the potential for developing SARDs and novel inhibitors, such as PROTACs, to overcome drug resistance in patients with PCa in the advanced and metastatic setting. 

Despite promising results for both modalities mentioned above, further studies and clinical trials would be needed for these novel approaches to be fully utilized in the everyday clinical setting. The development of novel pharmaceuticals based on the accumulated information regarding AR will ultimately reshape future treatment paradigms and clinical decisions in the ongoing treatment advancement for metastatic and advanced PCa.

## 8. Conclusions

Due to the advances of modern medicine, improvements in traditional therapies and novel treatments have evolved for the treatment of PCa and the therapeutic armory is continuously increasing. AR signaling is a crucial factor involved in the growth and progression of PCa. Therefore, in accordance with this disease spectrum, anti-AR molecules are in development to target the whole disease spectrum of PCa, from high-risk tumors to CRPC in the metastatic setting. However, there is a need for novel therapeutic approaches to improve the overall outcomes in patients with PCa. Many therapeutic agents with different AR-inhibiting mechanisms have shown efficacy in various clinical trials. Moreover, novel AR inhibitors, such as PROTACs and SARDs, have shown promising preclinical efficacy and safety profiles, therefore providing another option for overcoming resistance to current androgen-targeted therapeutics. Further prospective studies and accumulation of data from ongoing clinical trials will provide elucidation of the resistance mechanism of prostate tumors and the underlying molecular pathways for better treatment response.

## Figures and Tables

**Table 1 biomolecules-11-00492-t001:** Clinical trials evaluating androgen receptor inhibitor therapy for non-metastatic CRPC.

Agents	Clinical Phase	Identifier	Indication	Primary Endpoints
Enzalutamide	III	NCT00974311 (AFFIRM) [8]	CRPC	OS
Enzalutamide	III	NCT01212991 (PREVAIL) [10]	Chemotherapy-naïve mCRPC	OS and rPFS
Enzalutamide	III	NCT02003924 (PROSPER) [91]	Non-metastatic CRPC	MFS
Apalutamide	III	NCT01946204 (SPARTAN) [92]	Non-metastatic CRPC	MFS
Darolutamide	III	NCT02200614 (ARAMIS) [94]	Non-metastatic CRPC	MFS
Enzalutamide + Darolutamide	II	NCT03314324 (ODENZA) [97]	Asymptomatic or mildly symptomatic mCRPC	Patient preference
Enzalutamide + Darolutamide	II	NCT04335682 (ARACOG) [98]	mCRPC or non-metastatic CRPC	% change in the cognitive domain

CRPC: castration-resistant prostate cancer; mCRPC: metastatic castration-resistant prostate cancer; OS: overall survival; rPFS: radiographic progression-free survival; MFS: metastasis-free survival.

**Table 2 biomolecules-11-00492-t002:** Clinical trials evaluating androgen receptor inhibitor therapy for mHSPC.

Agents	Clinical Phase	Identifier	Indication	Primary Endpoints
Abiraterone acetate + ADT	III	NCT01715285 (LATITUDE) [5]	mHSPC	PFS and OS
Enzalutamide + ADT	III	NCT02677896 (ARCHES) [2]	mHSPC	rPFS
Enzalutamide + ADT	III	NCT02446405 (ENZAMET) [4]	mHSPC	OS
Apalutamide + ADT	III	NCT02489318 (TITAN) [3]	mHSPC	rPFS and OS

ADT: androgen-deprivation therapy; AE: adverse event; mHSPC: metastatic hormone-sensitive prostate cancer; OS: overall survival; PFS: progression-free survival; rPFS: radiographic progression-free survival.

**Table 3 biomolecules-11-00492-t003:** Clinical trials evaluating AR degradation therapy for the treatment of PCa.

Agents	Clinical Phase	Identifier	Indication	Primary Endpoints
ARV-110	I	NCT03888612 [115]	mCRPC	Dose-limiting toxicity, AE, PSA response

AE: adverse event; mCRPC: metastatic castration-resistant prostate cancer; PSA: prostate-specific antigen.

## Data Availability

Data sharing not applicable.

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
