# Peer review of "Current Status and Future Perspectives of Androgen Receptor Inhibition Therapy for Prostate Cancer: A Comprehensive Review"

_biomolecules, 2021, doi:10.3390/biom11040492_

Round 1

Reviewer 1 Report

The present review on current status and future perspectives of androgen receptor inhibition therapy for prostate cancer is interesting.

However, in the abstract authors stated that "In this review, the interaction between AR signaling and other molecular pathways involved in tumor pathogenesis and its clinical implications in metastasis and advanced disease will be discussed...", but they did not intensively developed this aspect in their review.

Another remark refers to the citation of the study reported in reference 27. In the first paragraph, page 3/17, authors mentioned "... malignant prostatic epithelial cells show lower levels of AR expression". But, in their study, Olapade-Olaopa et al., used immunohistochemical techniques to determine hAR positivity in the epithelium and adjacent stroma of sections from benign and malignant prostatic glands. AR expression has been evaluated by determining a score. The final scores were classified as follows: 0, negative; 1–33%, weak expression; 34–66%, moderate expression; >66%, high/strong expression.

It could be more appropriated to mention "mean AR expression" as in their study Olapade-Olaopa et al. reported increased heterogeneity of hAR staining in malignant prostatic epithelium compared to normal/atrophic and BPH glands in BPH.

Some references are required to strengthen the sentence : " In the disease spectrum of PCa, patients with metastatic disease have shown decreased AR stromal expression compared with the corresponding primary prostate tumor." (First paragraph, page 3/17).

References 35, 36, and 37 could be completed by reference 41 and doi: 10.1002/humu.20848.

Author Response

The present review on current status and future perspectives of androgen receptor inhibition therapy for prostate cancer is interesting.

REPLY: We are very much thankful to the reviewer for the thorough review. We agree to all specific comments addressed and have revised our paper in light of the useful suggestions. Answers to the specific comments/suggestions/queries are as follows.

However, in the abstract authors stated that "In this review, the interaction between AR signaling and other molecular pathways involved in tumor pathogenesis and its clinical implications in metastasis and advanced disease will be discussed...", but they did not intensively developed this aspect in their review.

REPLY: We agree that our previous manuscript did not detail the molecular aspects of prostate cancer (PCa) tumorigenesis. Clearly, it would be important to discuss the important mechanisms relevant to therapy failure. In accordance with the reviewer’s comment, we have added to ‘The androgen receptor and tumor microenvironment in prostate cancer (Section 2)’ information on cancer-associated fibroblasts (CAFs) which have been shown to be an important driver of PCa progression, the immune cell milieu including tumor-associated macrophages (TAMs) or alterations to the extracellular matrix which also play a role in PCa development and progression. Indeed, a better understanding of the underlying mechanisms will lead to multidisciplinary studies, which could provide some guidance for basic and translational research in the future.

Another remark refers to the citation of the study reported in reference 27. In the first paragraph, page 3/17, authors mentioned "... malignant prostatic epithelial cells show lower levels of AR expression". But, in their study, Olapade-Olaopa et al., used immunohistochemical techniques to determine hAR positivity in the epithelium and adjacent stroma of sections from benign and malignant prostatic glands. AR expression has been evaluated by determining a score. The final scores were classified as follows: 0, negative; 1–33%, weak expression; 34–66%, moderate expression; >66%, high/strong expression. It could be more appropriated to mention "mean AR expression" as in their study Olapade-Olaopa et al. reported increased heterogeneity of hAR staining in malignant prostatic epithelium compared to normal/atrophic and BPH glands in BPH.

REPLY: We agree that the information was described inappropriately. According to your comment, we have edited this sentence to: “starting from low-grade PCa, there is a linear decrease in mean AR stromal expression, which is almost entirely absent in high-grade disease. It has been shown that the mean AR expression is decreased both in the epithelium and adjacent stroma along with tissue dedifferentiation, while the depletion is more pronounced in the stromal nuclei.”

Some references are required to strengthen the sentence: “In the disease spectrum of PCa, patients with metastatic disease have shown decreased AR stromal expression compared with the corresponding primary prostate tumor." (First paragraph, page 3/17).

REPLY: Thank you for pointing this out. We have additionally cited the following reference for the abovementioned sentence: (Li Y. et al. J Cell Mol Med 2008, 12, 2790-2798).

References 35, 36, and 37 could be completed by reference 41 and doi: 10.1002/humu.20848.

REPLY: The aforementioned articles were cited as references 46 and 47 in the ‘Androgen receptor resistance mechanisms in treatments for advanced and metastatic prostate cancer’ section. We have edited our manuscript accordingly and have marked the changes.

Reviewer 2 Report

Drs. Kim and Koo have written a nice summary of the clinical trials examining current anti-androgens in combination with other agents to treat both early and late prostate cancer.

Comments:

  1. The authors say, "In this review, the intricate relationship between AR signaling and other molecular cascades involved with the pathogenesis of PCa and its therapeutic implications in advanced disease will be discussed, along with current and novel therapeutic approaches utilizing the AR signaling pathway." However, the review is really only focused on the latter (clinical trials), there is no discussion of interactions between AR signaling and other molecular cascades in any great depth. Thus this statement needs to be modified.
  2. I would recommend either removing the section titled "The androgen receptor and tumor microenvironment in prostate cancer" or completely rewriting it, as it really does not discuss the microenvironment and only cursorily talks about stroma. There is no mention of cancer associated fibroblasts (CAFs), which have been shown to be an important driver of PCa progression, or the immune cell milieu, including importantly tumor associated macrophages (TAMs), or alterations to the extracellular matrix which also play a role in PCa progression. 
  3. The Discussion section sums up the clinical trial information but doesn't discuss how these combination therapies might address the issues of AR resistance to therapy that were highlighted in section 3 ("Androgen receptor resistance mechanisms to treatments for advanced and metastatic PCa"). An expanded Discussion would enhance this review.

Author Response

Drs. Kim and Koo have written a nice summary of the clinical trials examining current anti-androgens in combination with other agents to treat both early and late prostate cancer.

REPLY: We are very much thankful to the reviewer for the thorough review. We agree to all specific comments addressed and have revised our paper in light of the useful suggestions. Answers to the specific comments/suggestions/queries are as follows.

The authors say, "In this review, the intricate relationship between AR signaling and other molecular cascades involved with the pathogenesis of PCa and its therapeutic implications in advanced disease will be discussed, along with current and novel therapeutic approaches utilizing the AR signaling pathway." However, the review is really only focused on the latter (clinical trials), there is no discussion of interactions between AR signaling and other molecular cascades in any great depth. Thus this statement needs to be modified.

REPLY: We agree that our previous manuscript did not detail the molecular aspects of prostate cancer (PCa) tumorigenesis. Clearly, it would be important to discuss the important mechanisms relevant to therapy failure. In regard to your current and next comment, we have added to ‘The androgen receptor and tumor microenvironment in prostate cancer (Section 2)’ more information on AR signaling and other molecular cascades involved in the development and progression of prostate cancer. We have also modified the corresponding state in the Introduction Section as follows: “The need to understand the intricate relationship between AR signaling and other molecular cascades involved with the pathogenesis of PCa and its therapeutic implications in advanced disease is of utmost importance, along with the familiarity of current and novel therapeutic approaches utilizing the AR signaling pathway for the treatment and management for patients with PCa in the advanced setting.”

I would recommend either removing the section titled "The androgen receptor and tumor microenvironment in prostate cancer" or completely rewriting it, as it really does not discuss the microenvironment and only cursorily talks about stroma. There is no mention of cancer associated fibroblasts (CAFs), which have been shown to be an important driver of PCa progression, or the immune cell milieu, including importantly tumor associated macrophages (TAMs), or alterations to the extracellular matrix which also play a role in PCa progression.

REPLY: Thank you for pointing this out. We agree that the aforementioned section does not comprehensively cover information on the tumor microenvironment (TME) and its role in the development and progression of prostate cancer (PCa). A major feature of PCa is a reaction in which cancer cells invade the stroma transforming the microenvironment into a background that is favorable to tumor development. Tumor cells within the stroma activate cancer-associated fibroblasts (CAFs), recruit inflammatory cells, transform the extracellular matrix (ECM) and release ECM-bound growth factors. Tumor-associated macrophages (TAMs) form the majority of cells in the TME and are usually activated and recruited by chemokines. TAMs have been found to activate CCL2, an inflammation-promoting chemokine that induces STAT3-mediated epithelial-to-mesenchymal transition (EMT). Activation of the EMT has been considered to play a key role in affecting the function of immune cells in the TME, contributing to immunosuppression and resistance. In light of your suggestion, we have edited ‘the androgen receptor and tumor microenvironment in prostate cancer’ section of our discussion to include additional information regarding the relationship between the AR and TME.

The Discussion section sums up the clinical trial information but doesn't discuss how these combination therapies might address the issues of AR resistance to therapy that were highlighted in section 3 ("Androgen receptor resistance mechanisms to treatments for advanced and metastatic PCa"). An expanded Discussion would enhance this review.

REPLY: We agree that there was an incomplete discussion on how the combination therapies might address the issues of androgen receptor (AR) resistance to therapy. In response to your suggestion and comments, we have added another section in our paper under the heading of ‘Future directions and perspectives in androgen receptor inhibitor therapy (Section 7)’. Tremendous progress has been made in elucidating the mechanisms of prostate cancer (PCa). These insights led to drug discovery efforts that ultimately expanded the therapeutic armory and dramatically improved the survival of men with PCa. Despite this success, many unmet needs remain, while the AR is considered as the key therapeutic target in PCa at all stages. Therapy resistance can arise via alterations in the AR, but new drugs and combination strategies can overcome this molecular challenge. The development of effective new drugs relies on detailed knowledge of the mechanism of action of AR, its structure, and how this changes when bound to ligands. We anticipate novel therapeutic approaches targeting different regions of the AR protein in the near future.

Round 2

Reviewer 2 Report

The authors did an excellent job addressing all comments and concerns in their revision.